# Biological Hazards and Indicators Found in Products of Animal Origin in Cambodia from 2000 to 2022: A Systematic Review

**DOI:** 10.3390/ijerph21121621

**Published:** 2024-12-03

**Authors:** Shwe Phue San, Rortana Chea, Delia Grace, Kristina Roesel, Sothyra Tum, Stephen Young, Tumnoon Charaslertrangsi, Nazanin Zand, Shetty Seetharama Thombathu, Ra Thorng, Leab Kong, Kuok Fidero, Linda Nicolaides

**Affiliations:** 1Natural Resources Institute, University of Greenwich, Medway ME4 4TB, UK; s.san@greenwich.ac.uk (S.P.S.); stephen.young@greenwich.ac.uk (S.Y.);; 2National Animal Health and Production Research Institute, General Directorate of Animal Health and Production, Phnom Penh 120603, Cambodia; 3International Livestock Research Institute, Nairobi 00100, Kenya; 4Science Division, Mahidol University International College, Nakhon Pathom 73170, Thailand; 5United Nations Industrial Development Organization, Phnom Penh 120101, Cambodia; t.shetty@unido.org (S.S.T.); l.kong@unido.org (L.K.); 6Ministry of Industry, Science, Technology, and Innovation, Phnom Penh 120203, Cambodia; kuok.fidero@misti.gov.kh

**Keywords:** biological hazards and hazard indicators, products of animal origin, systematic review, Cambodia, food control system

## Abstract

Biological hazards in products of animal origin pose a significant threat to human health. In Cambodia, there are few comprehensive data and information on the causes of foodborne diseases or risks. To date, there has been no known published study similar to this review. This systematic review is aimed to investigate the prevalence of biological hazards and their indicators in products of animal origin from 2000 to 2022. The main objective of this study was also to contribute to strengthening Cambodia’s food control system. This review followed the established “Preferred Reporting Items for Systematic Reviews and Meta-Analyses” (PRISMA) guidelines. In total, 46 studies were retained for complete review. Most studies (*n* = 40) had been conducted by or with external researchers, reflecting the under-resourcing of the National Food Control System in terms of surveillance; areas outside the capital were relatively understudied, reflecting evidence found in Ethiopia and Burkina Faso. Five categories of hazards were reported with the highest number of studies on fish parasites. Marketed fish, often originating from different countries, had a higher mean value of parasite prevalence (58.85%) than wild-caught fish (16.46%). Viral pathogens in bat meat presented a potential spillover risk. Many potentially important hazards had not yet been studied or reported (e.g., Norovirus, *Shigella*, toxin-producing *Escherichia coli*, and *Vibrio cholerae*). The findings of our review highlighted significant urgencies for national competent authorities to enhance food hygiene practices along the production chain, tackle import control, and enforce the implementation of a traceability system, alongside more research collaboration with neighboring countries and key trading partners. It is crucial to conduct more extensive research on food safety risk analysis, focusing on the identification and understanding of various biological hazards and their associated risk factors in food.

## 1. Introduction

Foodborne diseases (FBDs) are a critical concern on a global scale, as contaminated food can lead to more than 200 health issues, such as gastrointestinal, gynecological, and immunological disorders, as well as cancer. The majority of these illnesses can be effectively prevented [1]. Safeguarding food safety is a joint responsibility among various national authorities and requires an integrated, multisectoral approach that aligns with the principles of the one-health approach [2]. Regions in Africa and Southeast Asia bear the highest burden (more than 90%) of foodborne diseases [3,4]. The burden of FBDs in low- and middle-income countries (LMICs) comes from biological hazards, especially bacteria, parasites, viruses, and biotoxins, contaminating perishable food products mostly accessed from traditional markets [5,6]. Many food poisoning cases are attributed to the consumption of products of animal origin (POAOs) [7]. The leading causes of FBDs are primarily biological hazards, especially norovirus and *Campylobacter* spp. Furthermore, non-typhoidal *Salmonella enterica* followed by *Salmonella* Typhi, *Taenia solium*, and the hepatitis A virus are major factors in the fatalities associated with FBDs [8].

The role of the National Food Control System (NFCS) is crucial in safeguarding consumer health and promoting fair practices in food trade [9]. Nevertheless, the ability to effectively handle POAOs safely is generally weak in LMICs due to the limited ability to comply with food safety regulations and hygiene practices. Moreover, foodborne outbreaks in LMICs are often underestimated or not properly recorded because of the lack of sustainable surveillance systems [5,8,10]. The safety of POAOs is becoming a more significant concern due to higher levels of food consumption and longer, more complex supply chains that can lead to contamination by various hazards, including bacteria and parasites [11,12,13,14,15].

In the Kingdom of Cambodia (Cambodia), diarrhea accounts for approximately 8% of the mortality among children under the age of five and is linked to the ingestion of unsafe food. However, the specific causal agents of most diarrheal cases remain unidentified [16]. Between 2014 and 2023, the Cambodia Communicable Disease Control (C-CDC) reported a total of 178 incidents, resulting in 7224 cases and 180 fatalities attributed to FBDs. Many outbreaks have been primarily associated with the consumption of contaminated fish noodles and naturally toxic pufferfish [17]. There are also instances when the identification of hazards and the investigation of their causal agents remain incomplete [18].

The Cambodian NFCS is coordinated by the Council of Agriculture and Rural Development (CARD) of Cambodia, alongside six different line ministries, known as the Food Safety Working Group (FSWG), and other sub-committees, including the Foodborne Outbreak Report Team (FORT) which has the obligation to response to FBD outbreaks. The Ministry of Agriculture, Forestry, and Fisheries (MAFF) is mainly responsible for overseeing the safety of the primary production of POAOs [19]. Cambodia currently lacks a well-established surveillance system for foodborne pathogens, which consequently hinders the availability of comprehensive data. Nonetheless, an event-based surveillance system is in operation to collect reports related to public health events from a wide range of sources, including media channels and the public, who can report incidents via a dedicated hotline by FORT [17]. To date, there is no known comprehensive systematic review that examines the evidence of foodborne biological hazards associated with POAOs in Cambodia, especially in the context of bolstering the NFCS.

Our review aims to access the quantity of biological hazards (bacteria, biogenic amines, biotoxins, parasites, toxin-producing fungi, and viruses) and any associated information (type of study initiative, year of publication, food source, stage of value chain, and location) on POAOs in Cambodia over a period of 22 years, with the intention of gathering data and identifying the existing gaps to enhance the NFCS. The specific objectives are (1) to identify foodborne biological hazards and their indicators as detected in POAOs; (2) assess the quantity of contamination of biological hazards which have been found to exceed the Cambodian, Codex Alimentarius Commission, and European Union standards/recommended limits; (3) evaluate whether there is any association between the presence of a hazard and the level of hazard with the type of value chain and its inherent practices; and (4) reveal evidence of biological hazards reported in different provinces.

## 2. Materials and Methods

### 2.1. Protocol Development and Registration

The review protocol was developed based on the Preferred Reporting Items for Systematic Review and Meta-Analysis protocols (PRISMA-P) 2015 statement [20]. The concept of the protocol followed the previous review studies conducted by the International Livestock Research Institute (ILRI) for the Feed the Future Initiatives of USAID [21]. The protocol was registered to PROSPERO in March 2023.

The registration number PROSPERO 2023 CRD42023409476 can be found at the following weblink: https://www.crd.york.ac.uk/prospero/display_record.php?ID=CRD42023409476 (accessed on 27 March 2023).

The definitions of the key terms used in this review were listed in Appendix A.

### 2.2. Eligibility Criteria

The methods of this review followed the established “PRISMA” guidelines updated in 2020 [22].

#### 2.2.1. Inclusion Criteria

The inclusion criteria of the review were studies in English with a timeline from 1 January 2000 to 31 December 2022. The types of studies included were observational studies and reviews. The studies stated the prevalence and related information such as sampling and testing methods, and stages of the production chain of animals and POAOs were included.

#### 2.2.2. Exclusion Criteria

The exclusion criteria were studies in any other languages such as Khmer and studies focusing on non-foodborne hazards. Additionally, laboratory-based antimicrobial resistance-related studies without any information on sampling locations, sample numbers, analytical methods, and prevalence as well as the population outside Cambodia and studies with no prevalence data were not included in the review. Furthermore, studies of products not of animal origins were excluded.

### 2.3. Databases and Search Strategy

The search databases are Scopus, PubMed, and Google Scholar. The study selection included “(Foodborne OR “food borne” OR food-borne OR “food safety” OR “food related” OR “food associated” OR “food derived” OR “food* illness” OR “food* disease*” OR “food* intoxica*” OR “food pathogen” OR “food* poison*” OR “food* microb*” OR “food* vir*” OR “food parasit*” OR “food* toxin” OR “food* contamina*” OR “food* hazard*” AND (Cambodia*))”. Boolean operators (AND, OR, NOT, or AND NOT) were used to combine or exclude the keywords in the search databases.

### 2.4. Screening and Study Selection

All the search results from the three databases were compiled in a single Excel sheet and duplicates of the studies were removed in Microsoft Excel. The publication titles and abstracts were screened based on the inclusion and exclusion criteria of the study protocol. The screening was solely conducted by the first author, Shwe Phue San, (S.P.S) with the guidance of the supervisory team and external contributors. Full paper reviews were carried out manually. Full papers linked to the accepted abstracts were sought and acquired.

### 2.5. Quality Assessment Criteria

For the quality control, each selected paper was assessed using the following four quality criteria questions (for details, see Table 1).

Is the study method scientifically sound?Is the laboratory method used for testing biological hazards appropriate?Are the descriptions of data analysis for key outputs (prevalence or concentration) accurate and precise?Are the results and findings clearly stated?

These quality assessment criteria were adapted from the previous systematic literature review (SLR) conducted by ILRI [10]. The studies were classified as “good”, “medium”, and “poor” and only good- and medium-rated studies were selected for data extraction (see Table 1). The selected studies were presented to the research team for review and feedback.

### 2.6. Data Extraction

Articles found to be of acceptable quality after the full-text screening were considered for data extraction. The population of interest for the review was biological hazards including biogenic amines and biotoxins detected in food-sourced animals and in POAOs at any stage of the production chain. The extracted data included the type of animal and POAO, type and name of biological hazard, prevalence- and concentration-related information (total number of samples, number of positive results, type and stage of the production chain, geographical location, sample size, sampling method, analytical method, and year of publication), and type of initiative (national initiative or initiatives of the international institutions or joint initiatives between national and international institutions).

### 2.7. Data Analysis

Findings were heterogeneous and thus were primarily conveyed through descriptive analysis. However, the findings of the different types of parasites found in fish and fishery products yielded sufficient data to conduct a statistical analysis. One-way ANOVA was calculated to see if there were significant differences between the mean values of the prevalence of parasites yielded from three different types of sampling points (nature, village, and market). “Nature” means the samples were taken from natural habitats such as lakes, rivers, and seashores, whereas “villages” refers to the samples collected from the villages where the stage of the chain can be end-consumers or small sellers. Finally, “market” means the samples were purchased from retail or wholesale markets. Tukey’s test, also known as the Honestly Significant Difference (HSD) test, was subsequently used to find the mean values that are significantly different from each other.

### 2.8. Calculation of DALY/Population of Cambodia

The calculation of Disability-Adjusted Life Years (DALY) per population in Cambodia was based on the estimation provided by the World Health Organization (WHO) [8]. The population of Cambodia, according to the World Bank data from 2022, was 16,767,842 [23].

## 3. Results

A total of 8291 records were obtained from the three search databases (Figure 1). Following the elimination of duplicates in Excel, the number of records available for screening was reduced to 8221. Subsequently, during the “title screening” phase, 7968 records were excluded as they were found to be unrelated to food safety hazards. Consequently, the abstracts of 253 records were assessed, leading to the exclusion of 80 reports that focused on human cases rather than animals or POAOs. Ultimately, a total of 46 records were thoroughly reviewed, while 117 records were excluded due to various reasons outlined in Figure 1. Appendix A shows the list of the records included in the review.

### 3.1. Different Types of Research Initiatives

As shown in Figure 2, only 6 out of the 46 studies were undertaken by national research institutions without the involvement of international partners in terms of funding or technical support. In contrast, partnerships between national and two or more international institutions led to 15 studies being carried out, with financial or technical assistance provided by these collaborators. Among the bilateral studies, those conducted with Thailand had the highest frequency, followed by Sweden. The joint studies primarily involved countries from Europe. Additionally, there was one study each conducted in collaboration with Australia and the WHO under joint initiatives. The Republic of Korea, the United States of America (USA), and the French Republic (France) conducted a total of 13 studies without any partnership or collaboration with local institutions. In our review, all the studies conducted by the research institutions in the Republic of Korea focused on parasite contamination in fishery products. Except for the Kingdom of Thailand (Thailand), there was no evidence of bilateral research collaboration involving Cambodia and the neighboring countries of Laos PDR and The Socialist Republic of VietNam (Vietnam) within the scope of our review. Similarly, we did not identify any bilateral research collaboration between Cambodia and the People’s Republic of China (China), even in light of the Cambodia–China free trade agreement established in 2020 [24].

### 3.2. Frequency of Studies for Different Types of Biological Hazards

As shown in Figure 3, half of the total number of studies conducted focused on parasites in fish and fishery products and other POAOs, making it the most extensively researched area. Nearly one-third of the studies on parasites were carried out by researchers from the Republic of Korea. Following but not closely were studies on bacterial hazards in POAOs. Viruses, biogenic amines, and biotoxins accounted for a smaller proportion of studies, with five, four, and three studies, respectively. Additionally, we did not identify any studies focusing on the concentration of toxin-producing fungi or the prevalence of Norovirus and hepatitis A viruses detected in POAOs. Additionally, none of the studies included in our review identified the pathogenic strains of *Escherichia coli* (*E. coli*). Consequently, the *E. coli* referenced in the review are considered to be interpreted as hygiene indicators.

### 3.3. Number of Publications from 2000 to 2022

As shown in Figure 4, food safety research studies in Cambodia have been steadily rising since 2000. Specifically, the research frequency concerning food biological hazards in animals and POAOs more than doubled between 2011 and 2015, reaching its peak from 2016 to 2020. The latest year in our study (2021–2022) constituted 21% of the total studies retrieved. Our review found a total of four studies related to parasites and viral hazards from 2000 to 2005, with two studies focusing on viruses and the other two on parasites. Although the investigation of antibiotic resistance in *Salmonella* (S.) and *Campylobacter* species in retail poultry began in 2011, most of the antibiotic resistance or susceptibility studies reviewed were conducted after the year 2015.

### 3.4. Number of Reviewed Studies in Each Province in Cambodia

Most studies *(n* = 22) were conducted in Phnom Penh, the capital of Cambodia. Despite having comparable population densities, the studies on foodborne hazards in Kandal, Prey Veng, and Siem Reap did not reflect the same level of research activity with 11 studies conducted in Kandal province and 4 records were documented for the province of Prey Veng (Figure 5).

### 3.5. Evidence of Biological Hazards Reported in POAOs in Cambodia

In addition to bacteria, parasites, and viruses, biogenic amines and biotoxins were reported in the studies reviewed. Foodborne bacteria such as *Brucella* species, *Campylobacter* species, *Clostridioides* (*Cl.*) *difficile*, *Salmonella* species, and *Vibrio* (*V.*) species were observed. While *Brucella* species and *Cl. difficile* were found to have a low prevalence, the remaining bacterial hazards were detected at high levels. Additionally, high levels of hygiene indicators like *E. coli* and *Staphylococcus* (*Staph*.) *aureus* were reported. The prevalence of *Salmonella* species and *Staph. aureus* was highlighted in the review as an indicator of potential hazards. Furthermore, the presence of astrovirus and Nipah viruses in bats was noted. The detection of hepatitis E virus in pigs and pork products was also reported. Lastly, various parasites were detected in cattle, buffalo, pigs, and fishery products in Cambodia.

A summary of the evidence reported by the 46 studies included in our review is provided as Appendix A.

### 3.6. Prevalence of Bacterial Hazards and Hazard Indicators in POAO

In this review, 12 out of 46 studies (26%) had evidence of bacterial hazard prevalence in food-sourced animals and POAOs in Cambodia from 2000 to 2022. In addition to this, hazard indicators such as *E. coli* in POAOs and the prevalence of pathogenic bacteria on cutting boards used for chicken and pork meat at traditional markets were included.

#### 3.6.1. *Brucella* spp.

Out of the 12 studies reviewed for bacterial hazards and hazard indicators, only one study focused on *Brucella* spp. in cattle and swine [25]. As part of an animal disease surveillance program, a total of 1141 serological samples were collected from slaughterhouses in Takeo province, Cambodia, and screened with commercial enzyme-linked immunosorbent assay (ELISA) test kits, and doubtful samples were tested by real-time Polymerase Chain Reaction (PCR). These samples included 477 from cattle and 664 from swine. The seroprevalence of *Brucella* spp. in cattle was found to be 0.2%, while in swine it was 0.15%.

#### 3.6.2. *Campylobacter* spp.

In the review, two studies provided information on the prevalence of *Campylobacter jejuni*, *Campylobacter coli*, and *Campylobacter lari* in livestock and meat samples collected from various farms in Kampong Cham, Battambang, and Kampot provinces, and Phnom Penh. The 1005 samples were taken from chickens, ducks, cattle, pigs, water buffalo, quail, pigeons, geese, and pork carcasses. The studies utilized culture methods, PCR methods, and the ISO 10272-1 requirement to detect the *Campylobacter* spp. The PCR was more sensitive in detecting *Campylobacter* spp. than the culture method [26]. Among the different livestock species, pigs exhibited the highest prevalence of *Campylobacter* spp., with a prevalence of 72%. This was followed by 56% in chickens and 24% in ducks [26]. On the other hand, another study revealed 80.9% of *Campylobacter* species in pork carcasses in Phnom Penh [27].

#### 3.6.3. *Clostridioides* (*Cl.*) *difficile*

One study included in our review reported the first evidence of the presence of *Cl. difficile* in smoked and dried freshwater fish, specifically from Battambang, Kampong Chhnang, and Kampong Cham in Cambodia. However, the samples obtained from Kampong Thom and Siem Reap provinces did not exhibit the presence of the bacteria. Out of the 25 samples collected directly from the markets in the five provinces, 4 were found to be positive for *Cl. difficile* and were resistant to Clindamycin upon testing. Furthermore, after undergoing molecular analysis, three out of the four positive samples revealed the presence of toxicity genes A and B; however, none of the samples exhibited the gene fraction associated with the binary toxin CDT [28].

#### 3.6.4. *Escherichia coli* (*E. coli*)

Four out of the forty-six articles of our review focused on determining the presence of *E. coli* in various food products such as fishery products, poultry, and pork. In total, 1327 samples were from slaughterhouses and markets located in Phnom Penh, Bantaey Meanchay, and Siem Reap provinces [29,30,31,32]. These samples included fecal samples, broiler rectal swabs, carcass swabs, chicken caeca, chicken neck skins, rinse water, chopping boards, fish, and fermented fish known as Prahok. The prevalence of *E. coli* varied from being undetected in fermented fish to as high as 89.4% in chickens. The analytical methods used in these studies included the Afnor validation method, Biorad-Rad 07/01-07/93 and BRD 0717-12/04, methods adapted from the U.S. Food and Drug Administration (USFDA)’s BAM, and the ISO 9308-1 method for *E. coli* detection.

#### 3.6.5. *Salmonella* spp.

The prevalence of *Salmonella* spp. in various food sources and processing sites in Cambodia had been investigated in three studies included in the review [27,32,33]. In a study, a total of 684 samples were collected from traditional markets across all 25 provinces of the entire Cambodia [33]. These samples included chicken meat, cutting boards used for chicken, pork, pork carcasses, and cutting boards used for pork. The analysis conducted following the ISO6579:2002 standard revealed a prevalence of 42.6% *Salmonella* spp. in chicken meat, 41.9% in cutting boards used for chicken, 45.1% in pork, and 30.6% in cutting boards used for pork. Similarly, another study reported 88.2% positive findings of *Salmonella* spp. in pork carcass samples collected from the markets in Phnom Penh [27]. On the other hand, a study focused on the prevalence of *Salmonella* spp. in a specific fishery product called Prahok at the processing sites located in Siem Reap [32]. They collected 28 samples and analyzed them using the BAM method of USFDA and reported a prevalence of 3.5% *Salmonella* spp. in fermented fish. These findings highlighted the presence of *Salmonella* in various food sources and processing sites in Cambodia, emphasizing the need for appropriate food safety measures to prevent the transmission of this pathogen to consumers. According to the microbiological criteria of the EU, *Salmonella* must be absent in 25 g samples, in accordance with the sampling requirements laid down in the regulation. [34].

##### *Salmonella* (*S.*) *enterica*

In the review, three studies revealed the prevalence of *S. enterica* in poultry, pork, and fishery products [29,35,36]. In total, 1299 samples were collected from fish and fishery products, poultry products, and pork meat at the slaughterhouses and markets in Phnom Penh, Banteay Meanchey, and Siem Reap provinces. These samples included fecal samples, broiler rectal swabs, carcass swabs, chicken caeca, chicken neck skins, rinse water, chopping boards, and fish. The analytical methods were molecular identification and standard method ISO6579:2002 (E) for the detection of *Salmonella* in food. The prevalence of *S. enterica* ranged from 6% in broiler chickens to 100% in pig carcass samples at slaughterhouses.

#### 3.6.6. *Staphylococcus* (*Staph*.) *aureus*

In the review, only one study examined the occurrence of *Staph. aureus* in various samples obtained from traditional markets across all 25 provinces in Cambodia [33]. A total of 532 samples were gathered, including those from chicken, cutting boards used for chicken, pork, and cutting boards used for pork. The samples underwent testing for the presence/absence and quantification of coagulase-positive staphylococci (CPS) in accordance with ISO 6888-1:1999. The prevalence of *Staph. aureus* was found to be 38.2% in chicken samples, 17.7% in cutting boards used for chicken, 28.9% in pork meat samples, and 11.3% in cutting boards used for pork.

#### 3.6.7. *Vibrio* (*V*.) Species

Only one study in the review revealed the potential *Vibrio* risks in fermented fishery products (Prahok) [32]. A total of 28 samples were gathered from the processing facilities in Siem Reap province and subjected to an examination to identify *Vibrio* species using the partial adaptation method outlined in the BAM. The two different *Vibrio* tests yielded conflicting outcomes regarding the presence of *Vibrio* spp. The CHROMagar Vibrio test by using a chromogenic medium suggested the potential existence of *V. parahaemolyticus*, *V. vulnificus*, *V. cholerae*, and *V. alginolyticus* based on distinct colony colors. However, the Thiosulfate–citrate–bile salts–sucrose (TCBS) agar test indicated a negative result for *Vibrio* in all the samples.

### 3.7. Evidence of Antimicrobial Resistance Genes in Several Bacteria Found in POAO

Six studies presented the high percentages of antibiotic resistance in *E. coli*, *Salmonella*, and *Campylobacter* species, as well as the emergence of extended-spectrum beta-lactamase (ESBL)-producing *S. enterica* in slaughterhouses, markets, and retail meats in Phnom Penh and Banteay Meanchey, Cambodia [27,29,30,31,35,36]. A total of 1798 samples were collected from feces, carcasses, rectal swabs, skin, rinsed water, and chopping boards, with food source animals including fish, pigs, pork, and chicken. In addition, *Cl. difficile* isolated from smoked and dried freshwater fish showed their resistance to antibiotic clindamycin [28].

### 3.8. Evidence of Parasitic Hazards in Product of Animal Origins

A total of 23 out of 46 studies in our review revealed the presence of different types of parasites in POAOs in Cambodia. Among these, three studies highlighted the prevalence of *Faciola* and *Sarcocystic* species in cattle and buffalo, one study identified the contamination of *Gnathostoma spinigerum* in edible frogs, five studies demonstrated the presence of different parasites in pigs or pork, and thirteen studies concentrated on the evidence of various parasites in fish and fishery products [37,38,39,40,41,42,43,44,45,46,47,48,49,50,51,52,53,54,55,56,57,58,59]. The parasites were identified morphologically including the use of electronic microscopy, ELISA method, and molecular methods.

#### 3.8.1. *Fasciola* spp.

The evidence of prevalence of *Faciola* spp. in cattle ranged from 5% to 20% [55]. A total of 2391 fecal samples were collected from villages in Kampong Speu and Pursat provinces. Individual nematode egg counts were performed on the fecal samples using the quantitative McMaster method with a sensitivity of 50 eggs per gram of feces (EPG). The identification of gastrointestinal nematode genera was based on the morphological analysis of third-stage larvae sourced from the coprocultures of pooled samples.

##### *Fasiola* *gigantica*

In our review, two studies reported positive findings of *Fasciola gigantica* in cattle and buffaloes. The prevalence of bovine fasciolosis in Cambodia posed a risk to approximately 28% of cattle and buffaloes [37]. The study revealed 11.4% positive results (160 out of 1046 samples) for *Fasciola gigantica*. The fecal samples were collected from 11 provinces, namely Kandal, Kratie, Takeo, Kampong Speu, Kampong Cham, Pursat, Battambang, Kampong Thom, Kampong Chhnang, Prey Veng, and Svay Rieng, and a modified version of the Balivet egg count technique was used for analysis. Notably, Kandal province showed the highest positive rate, reaching 56.8%. In addition, another study reported 16.37% positive results of *Fasciola gigantica* from 171 fecal samples collected from villages in Pursat province and analyzed by using the Modified Balivat Fasciola egg counting technique [38].

#### 3.8.2. *Gnathostoma spinigerum*

A study included in the review found out a significant proportion of edible frog (*Hoplobatrachus rugulosus*) samples obtained from the market in Phnom Penh were contaminated with a parasite known as *Gnathostoma spinigerum*, with a prevalence rate of 60% [45]. However, no traces of this parasite were detected in the 10 edible frog samples collected from Takeo province, as well as in the 34 snakehead fish samples taken from the markets in Phnom Penh, Takeo, and Pursat provinces. This highlights the variation in the prevalence of *Gnathostoma spinigerum* among different regions and species, emphasizing the importance of monitoring and controlling the spread of this parasite to ensure food safety and public health.

#### 3.8.3. *Sarcocystis* Species

One of our review studies presented a 100% prevalence of *Sarcocystis* species, namely *Sarcocystis heydorni* and *Sarcocystis cruzi*, in the cardiac tissues of both cattle and buffaloes. Eight samples were collected from the hearts of these animals in Siem Reap province. The samples were subjected to microscopic examination and the presence of foodborne zoonotic pathogens was confirmed using molecular methods [57].

#### 3.8.4. Parasites in Pig/Pork Meat

The evidence of various types of parasites in pig and pork meat was examined through a review of five studies [39,40,41,58,59]. Please see Table 2 below for details.

#### 3.8.5. Parasites in Fish and Fishery Products

A total of 14 papers included in the review examined the prevalence of different parasite types in fish and fishery products [42,43,44,45,46,47,48,49,50,51,52,53,54,56]. Over the past 22 years, more than 9709 samples have been analyzed to detect parasites in fishery products. These studies were conducted in 10 out of the 25 provinces in Cambodia, namely Phnom Penh, Pursat, Kampong Cham, Takeo, Kratie, Kandal, Steng Trung, Siem Reap, Kampong Thom, and Prey Veng. The samples were collected from various sources such as lakes, rivers, aquaculture sites, the sea, villages, and markets, encompassing both pre-harvest and post-harvest stages. The detection methods employed in these studies were both morphological and molecular techniques.

Figure 6 displays the mean prevalence of various types of parasites found in fish and fishery products. The prevalence of *Haplorchis pumilio* was found to have the highest mean value, reaching 70%. On the other hand, *Haplorchis yokogawei* had the lowest mean value of prevalence, which was recorded as 15.35%. Additionally, there were other parasites present, including *Pygidiopsis cambodiensis* n. sp., *Stellantchasmus falcatus*, *Gnathostoma spinigerum*, *Procerovum* sp., *Centrocestus formosanus*, *Artyfechinostomum malayanum*, *Echinostoma mekongi*, and *Angiostrongylus cantonensis*. These hazards were categorized under “other parasites” due to the limited sample size for each parasite, making it impractical to present their individual mean values.

The mean value of the prevalence of parasites in the samples collected from markets was found to be 58.85%, while the mean values for the samples taken from nature and villages were 16.46% and 38.45%, respectively. We found that the prevalence of zoonotic parasites in fishery products ranged from 0.25% to 60% in the samples taken from nature (lakes and rivers), from 10% to 90% in the samples taken from villages, and from 6.7% to 100% in the samples taken from markets. According to the one-way ANOVA test, *p* value 0.0067 was observed, and thus, the differences in the mean values were highly significant. Consequently, Tukey’s test was conducted to identify the specific differences between the individual mean values. The results indicated that the mean values of the “nature” and “market” samples were significantly different, whereas the mean value of the “villages” sample did not differ significantly from either the “nature” or “market” samples. The letter codes (compact letter display) show the results of Tukey post hoc multiple comparisons: bars with the same letter are not significantly different at the *p* = 0.05 level.

### 3.9. Evidence of Viral Hazards in POAO

From 2000 to 2022, a total of five studies were carried out in Cambodia, focusing on viruses found in bats and pigs were included in the review [60,61,62,63,64].

#### 3.9.1. Astrovirus

In our review, we included evidence from a study that reported the prevalence of astrovirus which was found to be over 5% in bat samples collected from farms in Ratanakiri, Stung Treng, and Prey Veng provinces [60]. In addition to fecal samples, rectal, oral, and tissue samples were also collected, and a semi-nested PCR method was used for the identification of astrovirus.

#### 3.9.2. Nipah Virus

The review included three studies that specifically examined the Nipah virus in bats [61,62,64]. A total of 5867 samples were gathered from serum and urine utilizing serological methods. The samples were collected from roosts located in Phnom Penh, Battambang, Kampong Cham, Kandal, Prey Veng, and Siem Reap Provinces. The highest prevalence of the Nipah virus was identified in samples taken from restaurants in Kampong Cham (11.5%), whereas the samples from the natural environment in Battambang and Kandal exhibited the lowest prevalence of less than 2%.

#### 3.9.3. Hepatitis E Virus

Included in the review was a study that provided evidence of the presence of genotype 1 hepatitis E virus in fecal and serum samples obtained from pig farms in Phnom Penh [63]. The study reported a positive finding rate of 12.15% out of the 181 samples after undergoing a molecular analysis.

### 3.10. Biogenic Amines

In the review, four studies revealed different concentrations of biogenic amines in fish and fishery products in seven provinces of Cambodia, namely Phnom Penh, Kampong Som/Sihanouk Ville, Battambang, Kampong Chhnang, Kampong Cham, Kampong Thom, Kandal, and Siem Reap [65,66,67,68]. However, a study did not specify the exact location of the sampling point [65]. A total of 100 samples were collected from natural sources (such as lakes), fishponds, processing sites, cold storage facilities, and shops. The concentration of biogenic amines was determined and confirmed using advanced techniques including High-performance liquid chromatography with fluorescence detector (HPLC-FLD), Ultra-performance liquid chromatography (UPLC), and liquid chromatography–mass spectrometry (LC-MS).

Two studies reported low concentrations of histamine in freshwater fish ranging from “not detected” to 24.2 ppm [65,66]. Likewise, another study reported low levels of histamine in both freshwater and marine fishes ranging from 5.32 to 9.23 ppm [67]. On the other hand, the highest concentration of biogenic amines, particularly histamine (>500 ppm) and tyramine (>600 ppm), in different types of fermented fishery products were reported [68]. These findings exceeded both the Cambodian National limit of 100 ppm and the European Union limit of 200 ppm [34,69]. Except for fish sauce, there are no defined maximum limits (MLs) for histamine in other fishery products in Cambodia.

### 3.11. Biotoxins

Three studies that revealed the concentrations of paralytic shellfish toxins and tetrodotoxin in Mekong pufferfish and horseshoe crabs were included in our review [70,71,72]. A total of 49 samples were gathered from various locations including lakes, seashores, and wet markets in Phnom Penh, Kandal, Kratie, and Sihanouk Ville. The samples were analyzed by HPLC and LC-MS. The evidence of different concentrations of tetrodotoxin and paralytic shellfish toxins in horseshoe crabs and pufferfish is presented in Table 3.

### 3.12. The Estimates of Regional and National Foodborne Disease Burden in Cambodia

The Foodborne Disease Burden Epidemiology Reference Group (FERG), established by the World Health Organization (WHO), provided its initial findings on the global and regional impact of foodborne diseases. These findings included estimates of the occurrence, mortality, and overall burden caused by different foodborne hazards. As per the regional classification by WHO, Cambodia falls under the Western Pacific Region (WPR) B [8]. As shown in Table 4, the DALYs/population in Cambodia was calculated based on the DALYs/100,000 people in WPR B.

## 4. Discussion

As shown in Figure 2, our review included 46 studies, with only 13% being conducted by national research institutions. Additionally, 28% of the studies were carried out by researchers from the Republic of Korea, the French Republic, and the United States of America, without any collaborations with national research institutes. The remaining 59% of the studies were executed through either bilateral or multilateral partnerships with Australia, Austria, Belgium, Japan, Thailand, Sweden, WHO, and more than one international partner. Despite the considerably huge amount of food trade volume and trade agreements, we found no bilateral studies conducted between Cambodia and its primary food trading partners, particularly Vietnam and China [73]. From 2009 to 2018, the leading export partners for Cambodia were Vietnam, Thailand, China, Malaysia, and France. Annually, Cambodia imports around USD 1 billion in vegetables and meats, mainly sourced from Vietnam and Thailand [24].

The review found that 50% of the studies concentrated on foodborne parasites, which is particularly relevant to Cambodia, where helminths and cestodes are the predominant causes of the burden of FBD (Figure 3). Conversely, research on foodborne bacteria comprised 24% of the total studies. It highlights a significant gap in this area of biological hazards as 11 studies over 22 years are insufficient, especially considering the extensive variety of foodborne bacteria found in tropical climates like that of Cambodia. We discovered only five studies pertaining to viruses, specifically astrovirus and Nipah viruses in bats, and hepatitis E virus in pigs and pork. However, these studies may not represent the prevalence in the entire country because of the fragmented nature of the studies included in the review. Our search did not yield any significant foodborne viruses, including norovirus and rotavirus in fish, nor hepatitis virus in fish and meat. Additionally, we included four studies on biogenic amines in both freshwater and marine fish, as well as processed fishery products, and three studies on paralytic shellfish toxins and tetrodotoxin in pufferfish. Our review of the literature reveals a lack of studies specifically addressing biogenic amines, particularly histamine, in fish species like anchovy under the histidine-rich family Scombridae [74]. Instead, the existing research has predominantly concentrated on freshwater fish and groupers. Notably, one study did indicate the presence of low levels of histamine in mackerel, another species related to high histamine production.

As outlined in Figure 4, our review found a noticeable increase in the quantity of food safety research conducted after 2015 (Figure 4), suggesting a growing interest in this field of study. This increase in biological hazards over the specified timeframe is consistent with findings from another review that examined chemical and biological risks in urban agriculture and food safety systems in Global South countries. The review noted a substantial growth in the number of publications, reporting 39 articles during the decade from 2001 to 2010, and 123 articles in the subsequent decade from 2011 to 2020 [75]. The research conducted on biological hazards in products of animal origin (POAOs) during the years 2021 and 2022 constituted 20% of the total studies conducted over a span of 22 years, and this figure is anticipated to rise further by the conclusion of 2025 as several initiatives are being implemented in Cambodia with the objective of advancing food safety [18].

Like the patterns observed in Burkina Faso and Ethiopia [10,76], the studies were mainly concentrated in the capital city, Phnom Penh, as demonstrated in Figure 5. The evidence of the largest number of studies identified in this area is likely because of the highest population density in the country and the largest economic activities. In many LMICs, the capital city is responsible for a disproportionately high percentage of animal products consumed. In contrast, only one study has been executed in each of five out of the 25 provinces. The provinces of Pailin, Oddar Meanchey, and Mondulkiri are bordered by Thailand and the Lao People’s Democratic Republic, while the other two known as Koh Kong and Kap are coastal provinces. Thus, there is a significant requirement for additional research specifically targeting hazards in imported food and marine products.

The review highlighted various types of bacteria, specifically *Brucella* spp., *Campylobacter* spp., *Cl. difficile*, *Salmonella* spp., *S. enterica*, *Staph. aureus*, and *Vibrio* spp. which exhibited a broad host range including poultry, pigs, pork, cattle, buffalo, freshwater and marine fish, processed fishery products, edible frogs, and bats. Given the limited data on foodborne biological hazards in POAOs in Cambodia, we also incorporated hazard indicators such as *E. coli* as a hygiene marker, along with other biological contaminants found on chopping boards that come into direct contact with food.

The detection of the presence of *Staph. aureus* (11.3–38.2%) and *E. coli* (ND–89.5%) in fishery products, poultry, and pork indicated poor hygiene and sanitation practices along the food production chain in Cambodia. A study pinpointed that cross-contamination between the chopping boards used for meat is a potential source of biological hazards in traditional markets [33]. None of the studies that reported the prevalence of *E. coli* further tested the pathogenicity and toxicity. In contrast, an assessment of beef safety conducted in Egypt reported the prevalence (20–40%) of enteropathogenic and enterohaemorrhagic strains of *E. coli*. This study also reported a high prevalence of *Staph. aureus* (40–44%) and *E. coli* (68–80%) in imported or locally produced beef [77]. Likewise, a high prevalence of *Salmonella* spp. (3.5–88.2%) was observed in fish and fishery products, pork, and cutting boards. According to the EU’s microbiological criteria, *Salmonella* spp. must not be detected in 25 g of certain food such as fresh poultry meat [34]. A recent nationwide investigation in pig slaughterhouses in Thailand indicated that the non-compliance rates with the standards of the Department of Livestock Development were 22.34% for *E. coli*, 8.35% for *Staph. aureus*, and 30.10% for *Salmonella* spp. [78].

In a study, the prevalence of *Brucella* spp. in cattle and swine samples is reported to be low at 0.15%, while the pooled prevalence of Brucellosis across Asia stands at 8% [25,79]. However, the restricted number of studies complicates the ability to draw conclusive insights. We reviewed the first report on the occurrence of the pathogen *Cl. difficile* in ready-to-eat smoked and dried freshwater fish.. The implications of this finding emphasize the urgency for additional epidemiological studies to be conducted in this country [28]. Another study of our review indicated a significant prevalence of *Vibrio* species (98.2%) in processed fishery products in Siem Reap; however, additional verification is necessary due to the variability in the results obtained from different testing methods [32].

In our review, we found six studies that reported on antibiotic susceptibility, antimicrobial resistance (AMR), and the identification of extended-spectrum beta-lactamase (ESBL)-producing bacteria such as *E. coli*, *Salmonella*, and *Campylobacter* species in samples derived from poultry, fish, and pork products [27,29,30,31,35,36]. Furthermore, a study has reported the identification of resistance genes in *Cl. difficile* that are linked to resistance against clindamycin [28]. The prevalence of bacteria that were resistant to antimicrobials reported in these studies was alarming. The root causes of high prevalence were the absence of a strong legal framework relevant to the control of antibiotics use, weak enforcement of the responsible use of antibiotics, absence of GHP implementation, and lack of time and temperature management including inappropriate mode of transport for food products (e.g., meats were transported in open vehicles).

The review included a total of 23 studies that reported the prevalence of parasites in pigs/pork, fishery products, and ruminants in Cambodia. Alongside potential parasites such as *Opisthorchis* and *Trichinella*, several other parasites such as *Fasciola*, *Blastocystis*, *Ascaris*, and *Balantidium* species were included in the review because of the involvement of food-sourcing animals in the complex life cycles of these parasites. These parasites can be transmitted to humans not only through POAOs, but also through water, vegetables, and ready-to-eat food. According to the WHO’s estimate (Table 4), the highest burden of DALYs/population in Cambodia is due to helminth and cestode parasites.

As shown in Figure 7, the statistical analysis found that the prevalence of parasites in the samples collected from markets was significantly higher compared to those obtained from nature such as rivers and lakes. The fish sold at markets in Cambodia are mostly farmed fish and imported from the neighboring countries. The higher detection rates were very likely due to many factors such as inadequate import control [24] and the absence of good aquaculture practice (GAqP) implementation at fish farms. Furthermore, inadequate hygiene and sanitation practices may have contributed to an increased risk of cross-contamination especially from the intermediate host of trematodes such as snails and cats. In addition, the transportation and sale of live fish at markets could be contributing to the rise in prevalence; however, more research is needed to establish concrete scientific evidence. This indicates the necessity for the fishery competent authority to enforce a control mechanism such as recommending the specific freezing or heating time and temperature to kill parasites before consumption.

The evidence of parasites reported in the studies included *Opisthorchis (O.) viverrini*, *Haplorchis yokogawai*, *Happlorchis pumilio*, and other parasites as listed in Appendix A. *O. viverrini* is commonly known as small liver fluke and it can pose a serious public health concern in the Southeast Asian region. The finding of *O. viverini* in Cambodia was expected because the Greater Mekong Sub-region (GMS) is known as a high endemic area for *Opisthorchis* and opistorchiasis [80]. Not only humans, but also domestic animals can play a role in contaminating water sources with fecal eggs, which can then lead to infections in snails and fish. As a result, it is recommended that community-based surveys should be conducted in the near future to assess the prevalence of *O. viverini* in humans, domestic animals, and fish [51].

Freezing treatment for parasites, as mandated by EU legislation, requires the reduction of the temperature throughout the entire product to either −20 °C or below for a minimum duration of 24 h, or to −35 °C or below for a minimum duration of 15 h, to effectively control parasites other than trematodes. The European Food Safety Authority (EFSA) cites the WHO’s findings, which state that freezing at −10 °C for a period of 5 days is sufficient to eliminate the metacercaria of *Opisthorchis* spp. [81].

A study report emphasized the vulnerability of POAOs to spoilage and contamination during collection or slaughtering, despite their elevated protein levels. These difficulties are exacerbated in the absence of dependable cold chain management systems [18]. Likewise, our review results demonstrated a markedly higher parasite prevalence in the fishery product samples obtained from markets compared to those sourced from natural environments like lakes and rivers, suggesting an increase in the detection of parasites very likely because of complex and compounded factors such as the lack of good hygiene practices (GHPs) at fish farms, insufficient import control, and likelihood of cross-contamination during transport because fish sold at markets are mostly either farmed or imported from the neighboring countries.

The monitoring and surveillance of animal husbandry practices and antibiotic use in livestock in Cambodia, as well as in neighboring countries, are lacking. Antibiotic usage in food animals is often inappropriate and unregulated, with farmers prioritizing production benefits over the potential negative impacts of antibiotic use. To address this issue, a comprehensive program focusing on the responsible use of antibiotics in food animals should be implemented, starting with feed retailers and the commercial industry. All veterinary products should be labeled in Khmer to ensure proper understanding according to the requirements of the national regulations. Training programs for farmers should be conducted by an independent agency to promote knowledgeable and responsible antibiotic use [82].

In our review, we examined five studies that concentrated on three distinct foodborne viruses: astrovirus, hepatitis E virus, and Nipah virus. These studies provided evidence regarding their prevalence in pigs, pork, and bats. However, we found no studies that documented the presence of other significant foodborne viruses, particularly norovirus, rotavirus, and hepatitis A virus, in various meat and fishery products that are known to be major sources of foodborne viruses [83]. There is a pressing need for more comprehensive research on viruses, as the food safety management practices designed for bacteria and other microorganisms may not be effective against viruses. This necessity has become even more pronounced in the wake of the COVID-19 pandemic, highlighting the importance of understanding viruses that could impact public health.

The studies included in this review revealed the range of biogenic amines in fishery products from 8.1 to 2035 ppm. One out of four studies reported high levels of two biogenic amines (histamine > 500 ppm and tyramine > 600 ppm) which were above the national and EU limits for histamine in fishery products [68]. In our review, three out of four biogenic amine studies have investigated the histamine content in freshwater fish [65,67,68]. Even though freshwater fish are not typically classified as high histamine producers, it is vital to assess hygiene measures and the management of time and temperature throughout the production chain. Furthermore, due to the popularity of anchovy fish sauce in Cambodia, it is imperative that additional research on histamine levels in this product be conducted to protect consumer health. The production process, distribution, and domestic handling of fermented products in Cambodia should be re-evaluated to minimize the content of biogenic amines and microbiological contamination. Further research is required to establish preservation techniques that could be applied on an industrial level and small scale in Cambodia.

Between 2017 and 2019, the Ministry of Health in Cambodia reported a minimum of seven cases of poisoning resulting from the consumption of freshwater pufferfish. These incidents led to the pufferfish poisoning of over 40 individuals and five fatalities [72]. In our review, three studies reported the paralytic shellfish toxin and tetrodotoxin toxin in horseshoe crabs and pufferfish. Variations in toxin levels have been documented across various species of Cambodian freshwater pufferfish. Additional research is necessary to explore the specific types of toxins produced by each species of toxin-producing fish.

Many studies have examined the pathogens found in various food products, but most of them have focused on specific pathogens, for example, *Campylobacter jejuni*. These studies are limited in scope and only identify a few pathogens of interest that Cambodians may be exposed to through food. However, they do not necessarily identify the exact agents responsible for foodborne illnesses across the country. For instance, a food item may contain *Salmonella* which is not a zoonotic *Salmonella*, or it is not a *Salmonella* that can cause diarrhea, but the illness could be caused by other potential foodborne pathogens which were not specifically studied [16].

Notably, the review did not identify several important foodborne biological hazards in POAOs, including Norovirus, hepatitis A virus, *Bacillus cereus*, *Shigella*, *Listeria monocytogenes*, and toxin-producing *E. coli* in Cambodia. Despite the rise in research studies on foodborne hazards, there remains a significant disparity in understanding the risk profile of numerous potential foodborne hazards in Cambodia that could impact the safety of POAOs.

Enhancing food safety sensitization initiatives for individuals involved in the high-risk value-chain, enforcing and implementing comprehensive regulatory frameworks, amplifying communication with consumers and other relevant value chain actors regarding safe food practices, and upgrading national quality infrastructure, including clean water availability, washing stations, cold storage facilities, and logistics management, all have the potential to enhance food control system in Cambodia [84]. The Food Safety Law (2022) in Cambodia covers the supervision and guarantee of safety, quality, hygiene, and legality across all the phases of the food production process, as stated in Article 1. The Ministry of Commerce is empowered to take charge of and synchronize the governance, in collaboration with other ministries and institutions that are implicated in the sphere of food quality and safety, while the General Directorate of Customs and Excise (GDCE) is responsible for the control of food product import and export [19].

## 5. Conclusions

The food safety challenges already identified by the findings of our reviewed studies included risky consumption patterns such as eating raw or partially cooked POAOs, lack of awareness and implementation of good practices, weak law enforcement, absence of consistent sampling and testing, limited funding, and dependency on aid agencies. In addition, the recommendations of the previous studies entailed the need to establish effective monitoring and surveillance mechanisms, build up a strong and competent food testing network, and strengthen disease prevention and control plans in animal husbandry.

The findings of our review highlighted urgent requirements for effective strategies to strengthen the food control system in Cambodia. The Cambodian government is encouraged to initiate regional collaboration in food safety research and studies, alongside the legalization of a traceability system within the high-risk food production chain. Lastly, conducting a nationwide survey on food consumption is essential for the effective creation of a risk-based national food control system. In addition to fostering GHPs, it is vital to establish cold chain facilities and develop appropriate infrastructure for processing, storage, and transportation. In early 2024, the Fisheries Administration enacted two decisions to regulate VMP usage and to implement a National Residue Monitoring Plan (NRMP) for aquaculture products in Cambodia. Nevertheless, the priority for the MAFF is to control the misuse and abuse of veterinary medical products (VMPs) at the primary production level. Additionally, the GDCE in collaboration with technical ministries such as MAFF and the Ministry of Health should strengthen risk-based food import controls, particularly in relation to border trade with neighboring nations. Generally, border trade, often termed informal trade control, poses considerable challenges due to the high frequency of small food commodity movements across border checkpoints.

The evidence of our review suggested that the existing data and information on biological hazards in POAOs in Cambodia are not sufficient to draw a solid conclusion for the risk profile of each biological hazard in animals and POAOs because of the limited and unproportionate geographical coverage and absence of prevalence data for the agents that contribute to the highest DALYs in WPR B, such as toxin-producing fungi, toxin-producing *E. coli*, *Shigella*, and *V. cholerae*. Nevertheless, it is obvious that more research studies are required to better understand the risk factors and risk pathways of the biological hazards of POAOs in order of priority. There is a need for additional systematic reviews that follow PRISMA guidelines, exploring different types of hazards in high-risk food products aside from POAOs. This should include an examination of fresh fruits and vegetables, ready-to-eat foods, and the safety of drinking water and water used in food business operations.

The two limitations of this review were the possibility of selection bias because the screening of the studies and data extraction were solely carried out by the individual reviewer, and challenges in the interpretation of the results considering the heterogenicity of the findings. These limitations, however, were minimized by the contributions of the research project supervisors and external experts. Additionally, the geographical locations and names of provinces referenced in this study, particularly concerning the processed fishery products and other samples collected at markets, pertain solely to the sampling sites and do not necessarily indicate the actual production sites or sources.

## Figures and Tables

**Figure 1 ijerph-21-01621-f001:**
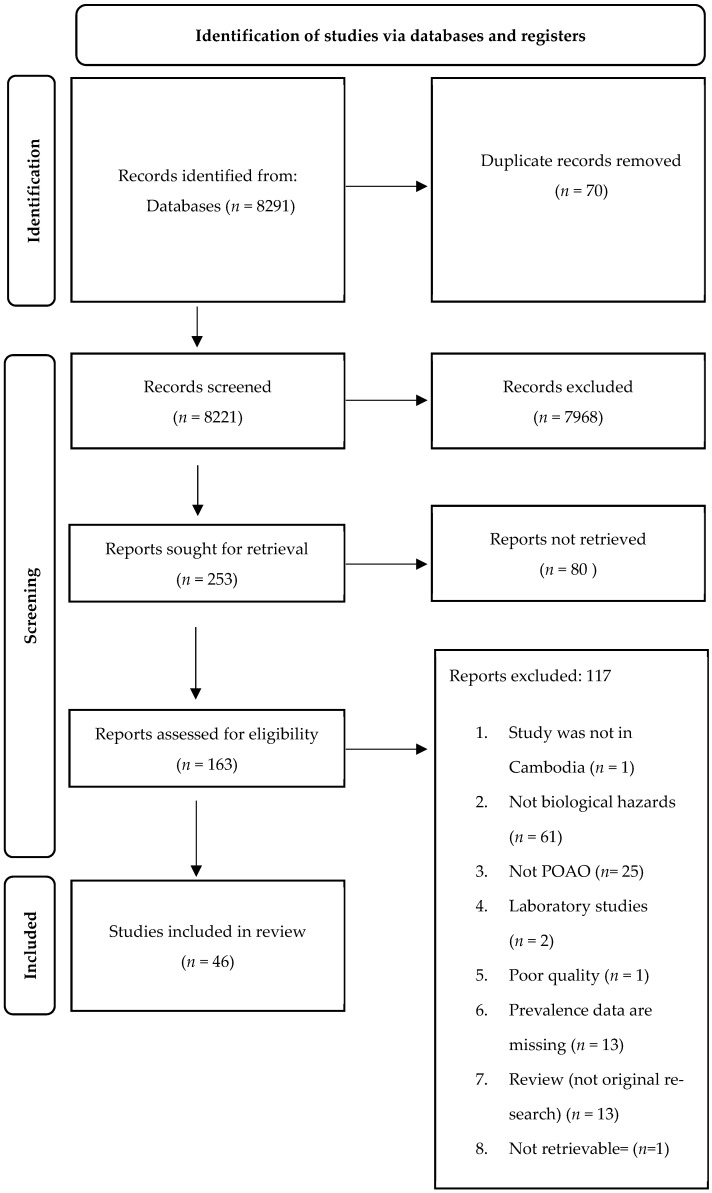
PRISMA flowchart showing identification, screening, and inclusion of eligible articles reporting foodborne biological hazards in animals and POAOs in Cambodia from 2000 to 2022.

**Figure 2 ijerph-21-01621-f002:**
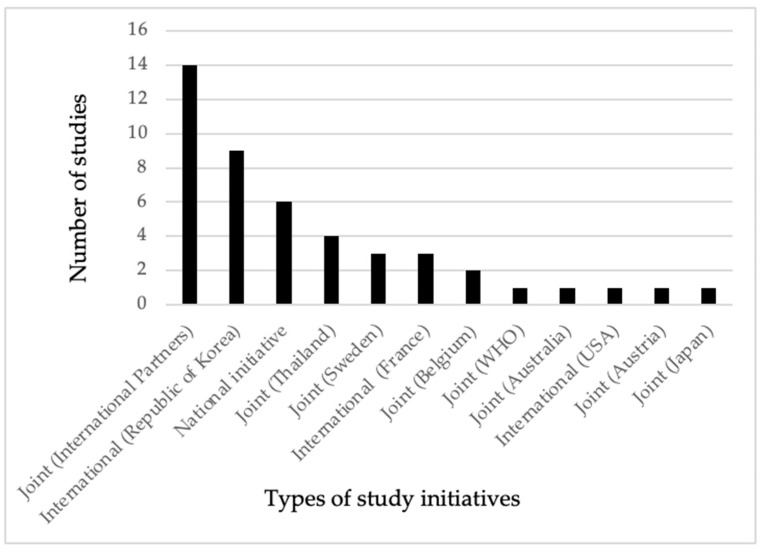
Evidence of frequency of research for food biological hazards in animals and POAOs conducted by national initiatives, joint initiatives, and international institutions in Cambodia (2000 to 2022).

**Figure 3 ijerph-21-01621-f003:**
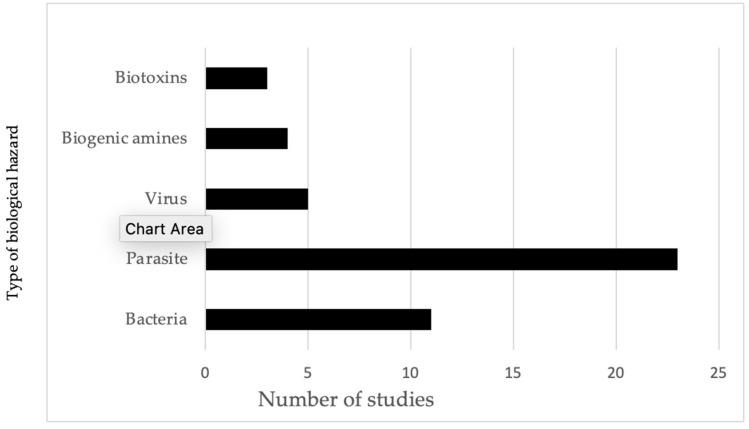
Evidence of frequency of studies identified for different types of biological hazards in animals and POAOs in Cambodia from 2000 to 2022.

**Figure 4 ijerph-21-01621-f004:**
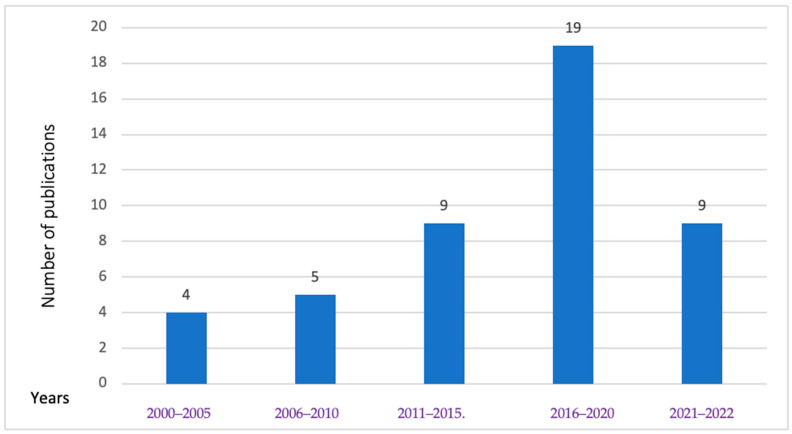
Number of publications of biological hazards in animals and POAOs in Cambodia for different intervals from 2000 to 2022.

**Figure 5 ijerph-21-01621-f005:**
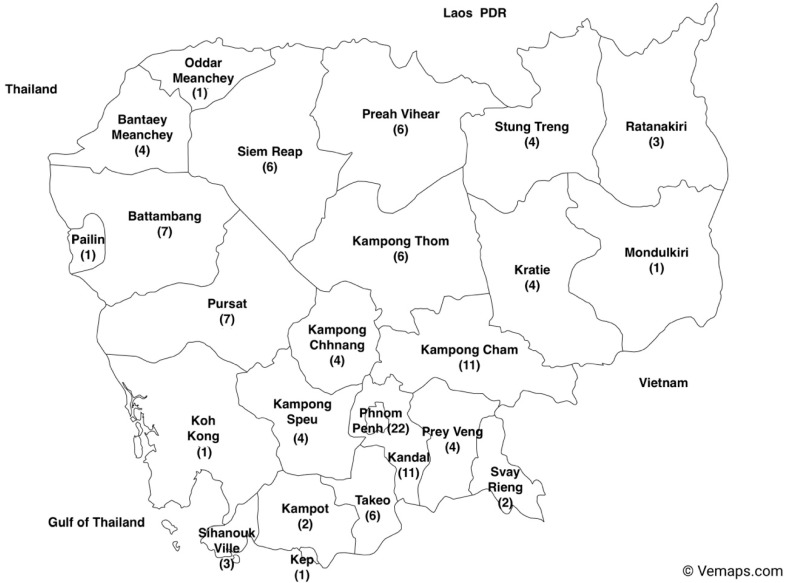
Number of studies included in the systematic review in each province in Cambodia from 2000 to 2022 (the map of Cambodia with provinces was downloaded from Vemaps.com (accessed on 17 December 2023).

**Figure 6 ijerph-21-01621-f006:**
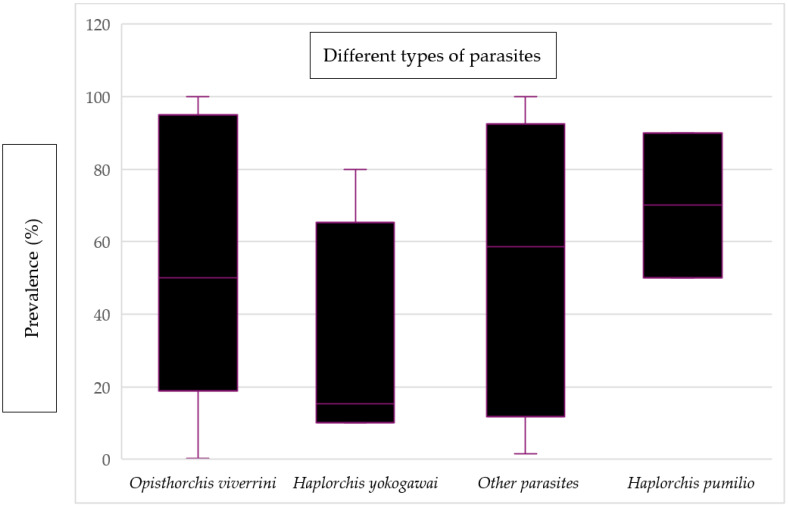
Evidence of the prevalence of different types of parasites in fish and fishery products.

**Figure 7 ijerph-21-01621-f007:**
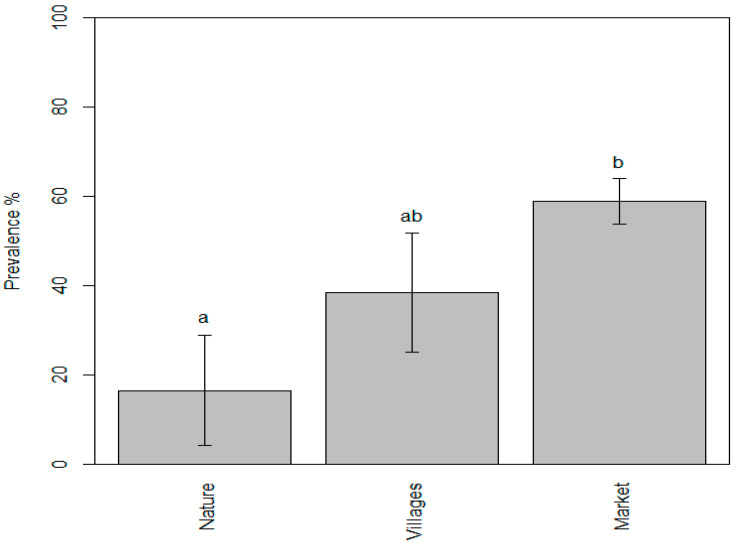
Evidence of parasite prevalence in fish and fishery products taken from three different points of sampling showing the significant difference between “nature” (a) and “market” (b).

**Table 1 ijerph-21-01621-t001:** Quality assessment criteria for full-text review.

Quality Criteria	Good	Medium	Poor
Scientific method	1. Clear description of study subject, study setting, and sampling points in detail 2. Appropriate sampling method	1. Detailed description of the study subject, study setting, and sampling points but somewhat unclear. 2. Sampling methods not well described but remain acceptable	1. Study subject, study setting, and sampling points are not described at all. 2. Unclear/invalid sampling methods
Testing method	1. Standard laboratory testing methods were used (use of relevant ISO method or equivalent)	1. The laboratory testing methods were acceptable or valid (use of the Bacteriological Analytical Manual (BAM) method or equivalent)	1. The laboratory testing method was not acceptable or was invalid (the use of rapid test kits).
Information for the key outputs	1. Detailed information available for data analysis	1. Sufficient information for data analysis	1. Insufficient or unclear information for data analysis
Results accuracy	1. Detailed and accurate results	1. Sufficient results for data extraction	1. Insufficient or incomplete result presentation

**Table 2 ijerph-21-01621-t002:** Evidence of different types of parasites in pigs and pork in Cambodia.

Scientific Name	Sampling Location/s	Sample No.	Sample Source	% of Positive Findings	Reference (Paper No. in Appendix A)
*Ascaris* spp.	Preah Vihear	76	Fecal	26.3	35
*Ascarops* spp.	Preah Vihear	76	Fecal	2.6	35
*Balantidium coli*	Preah Vihear	76	Fecal	15.8	35
*Blastocystis* spp.	Preah Vihear	73	Fecal	45.2	7
*Capillaria* spp.	Preah Vihear	76	Fecal	5.3	35
*Eimeria* spp.	Preah Vihear	76	Fecal	6.6	35
*Entamoeba* spp.	Preah Vihear	76	Fecal	31.6	35
*Gnathostoma doloresi*	Preah Vihear	76	Fecal	9.2	35
*Metastrongylus* spp.	Preah Vihear	76	Fecal	19.7	35
*Oesophagostomum* spp.	Preah Vihear	76	Fecal	73.7	35
*Taenia* spp.	Phnom Penh, Kampong Thom, Preah Vihear, Ratanakiri, and Stung Treng	242	Blood	0–31.4	30
*Taenia solium*	Phnom Penh, Kahdal, and Kampong Speu	1492	Blood	4.1–16.7	2, 28
*Trichinella* spp.	Kampong Thom, Preah Vihear, Ratanakiri, and Stung Treng	1114	Blood	1.5–4.9	2, 30
*Trichuris suis*	Preah Vihear	76	Fecal	19.7	35

**Table 3 ijerph-21-01621-t003:** Evidence of concentration of biotoxins in horseshoe crab and pufferfish in Cambodia from 2000 to 2022.

Value	Highest Value of Tetrodotoxin in Horseshoe Crab Mouse Units (MU)/g [70]	Average Concentration of Paralytic Shellfish Toxin in Organs of Freshwater Pufferfish (mg STXdi·HCL eq/kg) [72]	Level of Tetrodotoxin in Skin and Reproductive Organs of Mekong Pufferfish (MU)/g [71]
Minimum	38	0.5	4
Maximum	315	99.4	37
Range	277	98.9	33
Mean	102.83	22.74	14.8

**Table 4 ijerph-21-01621-t004:** Calculation of DALY/Population in Cambodia based on DALY/100,000 in WPR B region of WHO.

Biological Hazard	DALYs/100,000 People in WPR B	DALYs/16,767,842 (Cambodia)
Helminths	162	27,163.9
*Paragonimus* spp.	60	10,060.7
Cestodes	45	7545.5
*Salmonella* Typhi	36	6036.4
*Clonorchis sinensis*	31	5198.0
*Taenia solium*	27	4527.3
*Echinococcus multilocularis*	18	3018.2
Aflatoxin	17	2850.5
*Ascaris* spp.	11	1844.5
*Campylobacter* spp.	10	1676.8
*Shigella*	9	1509.1
*Toxoplasma gondii*	9	1509.1
*Salmonella Paratyphi* A	8	1341.4
Enteropathogenic *E. coli*—EPEC	5	838.4
Hepatitis A virus	5	838.4
Enterotoxigenic *E. coli*—ETEC	4	670.7
Non-typhoidal *S. enterica*	4	670.7
Norovirus	4	670.7
*Opisthorchis* spp.	3	503.0
*Listeria monocytogenes*	1	167.7
*Fasciola* spp.	0.9	150.9
*Brucella* spp.	0.6	100.6
*Cryptosporidium* spp.	0.3	50.3
*Entamoeba histolytica*	0.3	50.3
*Giardia* spp.	0.3	50.3
*Echinococcus granulosus*	0.3	50.3
*Vibrio cholerae*	0.1	16.8
Shiga toxin-producing *E. coli*	0.01	1.7
*Trichinella* spp.	0.004	0.7

## Data Availability

Data are contained within the article and Appendix A.

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
