# Peer review of "Biological Hazards and Indicators Found in Products of Animal Origin in Cambodia from 2000 to 2022: A Systematic Review"

_ijerph, 2024, doi:10.3390/ijerph21121621_

Round 1
Reviewer 1 Report
Comments and Suggestions for Authors
The manuscript was clearly written and the results were thoroughly discussed. I'm not sure if it is the type of journal, but I've noticed in multiple places that the source of the citation was not mentioned but a numbered citation was used directly. Here are the detailed comments:
Line 53: According to what? The information is missing.
Line 65: Salmonella should be italicized.
Line 84 and 330: Same comment as line 53, unless it’s this journal’s style.
In the results session, the order of biological hazards is confusing. It seems like it was organized by bacteria, parasites, etc. however, Sarcosystis is among all the bacteria. Having some titles indicating how the following paragraphs are grouped would be helpful.
Table 5: it is not figure 5. It would be a clearer comparison to use the same population scale.
Author Response
Comments from reviewer 1
Comment 1: The manuscript was clearly written and thoroughly discussed results. I'm not sure if it is the type of journal, but I've noticed in multiple places that the source of the citation was not mentioned but a numbered citation was used directly. Here are the detailed comments:
Response: Thank you for the positive feedback. A major revision is carried out based on the comments of all reviewers. A numbered citation is the referencing style of the journal, and the in-text citations are improved. Please see the revised manuscript.
Comment 2: Line 53: According to what? The information is missing.
Response: In-text citation style has been revised and now it is in line with the journal’s requirements. Please see line number 98.
Comment 3: Line 65: Salmonella should be italicized.
Response: Thank you. Please see the line number 101.
Comment 4: Line 84 and 330: Same comment as line 53, unless it’s this journal’s style.
Response: The in-text citation is revised according to the journal’s requirement. Please see lines 108.
Comment 5: In the results session, the order of biological hazards is confusing. It seems like it was organized by bacteria, parasites, etc. however, Sarcosystis is among all the bacteria. Having some titles indicating how the following paragraphs are grouped would be helpful.
Thank you very much for pointing this out. The order of Sarcocystic is re-arranged. The sub-headings are also re-arranged. Please see line 467.
Comment 6: Table 5: it is not figure 5. It would be a clearer comparison to use the same population scale.
Response: Thank you once again. The Table number has been rearranged, and the table caption has been improved. The table calculates the DALY/population of Cambodia based on the DALY/100,000 in WPR B. Please see line 554.

Reviewer 2 Report
Comments and Suggestions for Authors
1. The title should be concise
Abstract section
2. The abstract should be revised and be concise
3. Please avoid using abbreviations
4. What is the main objective of this study?
5. Authors should stress the novelty of this work
6. I invite authors to add some numerical data
7. A comparative data should be introduced
8. A 2-3 concise and conclusive sentences should be added at the end of the abstract
Introduction section
9. All references should be adjusted according to mdpi format
10. I advise authors to add a subsection ‘Abbreviations” after the abstract section, since several abbreviations were used here
11. This sections should be rewritten, it is very Long and please check English language,
12. GENERAL data should be avoided, please focus on the main matrix studied on this MS
13. I invite authors to use recent and proper references (2020-2025), and more sentences should be developed
14. According to 5, ???
15. Authors should stress the novelty of this work since several studies developed the same idea
16. L65-66, please develop more this sentence
17. L103, please add a suitable reference
18. More explain this terme: biological hazards (and related information)
19. The main objective was not clear, improve it
Materials and Methods section
20. the Material S1 was not provided
21. S.P.S, how about this abbreviation
22. What is corresponding criteria “s the laboratory method used for testing biological hazards appropriate?” EVALUATED
23. all tables should be in order, Table 1, Table 2, Table 3, ….be careful (Table 5 L 217)
Results section
24. the figure 2 and 4. should be more discussed
25. the Table 2 should be inserted as sup. File
26. In this paper, discussion needs to be improved. Results presented need a better discussion
27. There was no enough discussion or analysis of the results. The author should explain and clearly discuss this part based on scientific knowledge. It is better to compare the results with more similar recent works. And discuss the superiority of the work.
28. To further contextualize the significance of the research, can the authors discuss potential future applications? Additionally, it would be helpful to highlight any limitations of the research.
29. all data should be linked, since this study has several data
Author Response
Comments from reviewer 2
We thank you very much for very thorough review for improvements.
Comment 1: The title should be concise
Response: Thank you very much for the comments. The title has been revised, and we removed the repetitive words.
Abstract section
- The abstract should be revised and be concise
The abstract has been revised according to the comments.
- Please avoid using abbreviations
Thank you. We avoided the abbreviations for the first time and list of abbreviation is also separately added. We request to keep some important acronyms such as POAO and E coli.
- What is the main objective of this study?
The main objective is added. Please see the lines 21 and 22.
- Authors should stress the novelty of this work
Thank you. We added a sentence. Please see line 19.
- I invite authors to add some numerical data.
Thank you. The numerical data are added. Please see line 29.
- A comparative data should be introduced
Please see the lines 26 and 27.
- A 2-3 concise and conclusive sentences should be added at the end of the abstract
Thank you. We added lines 32 to 35.
Introduction section
- All references should be adjusted according to mdpi format
Thank you for pointing this out. We revised the references. Please see the revised manuscript.
- I advise authors to add a subsection ‘Abbreviations” after the abstract section, since several abbreviations were used here
We have added the Abbreviation subsection. Please see the line 42 to 87.
- This section should be rewritten, it is very Long and please check English language,
We have followed the advice. Please see the revised manuscript.
- GENERAL data should be avoided, please focus on the main matrix studied on this MS
Please see the revised introductory section.
- I invite authors to use recent and proper references (2020-2025), and more sentences should be developed
Thank you and the updated references are added. Please see the revised manuscript.
- According to 5, ???
The in-text citation is improved. Please see the line 98.
- Authors should stress the novelty of this work since several studies developed the same idea
We added a sentence. Please the lines 129-132.
- L65-66, please develop more this sentence
We replace the lines with more updated information. Please see the lines 99-102.
- L103, please add a suitable reference
Thank you. The reference is added. Please see line 117.
- More explain this terme: biological hazards (and related information)
The terms are elaborated. Please see the lines 133 to 136.
- The main objective was not clear, improve it.
The main objective is revised. Please see lines 133 to 137.
Materials and Methods section
- the Material S1 was not provided
The S1 is submitted together with the revised manuscript. Thank you.
- S.P.S, how about this abbreviation
Please see the line 184.
- What is corresponding criteria “s the laboratory method used for testing biological hazards appropriate?” EVALUATED
The corresponding criteria are elaborated in below table. Please see the third row of table 1 (under the line 201)
- all tables should be in order, Table 1, Table 2, Table 3, ….be careful (Table 5 L 217)
Thank you once again. The tables are now put in order. Please see the line 546.
Results section
- the figure 2 and 4. should be more discussed
The discussions are elaborated. Please see the lines 286 to 293 for figure 2 and lines 317-321 for figure 4.
- Table 2 should be inserted as sup.
Please see the supplementary material 3.
- In this paper, discussion needs to be improved. Results presented need a better discussion
Please see the entire discussion section of the revised manuscript. Thank you very much.
- There was no enough discussion or analysis of the results. The author should explain and discuss this part based on scientific knowledge. It is better to compare the results with more similar recent works. And discuss the superiority of the work.
Please see the entire discussion section of the revised manuscript. We added some comparison data. Please see lines 580-584, line 588, line 606, and line 610.
- To further contextualize the significance of the research, can the authors discuss potential future applications? Additionally, it would be helpful to highlight any limitations of the research.
Thank you for the suggestion. We added the potential future research. Please see lines 738-742.
- all data should be linked, since this study has several data
Thank you very much. Please see the revised manuscript.

Reviewer 3 Report
Comments and Suggestions for Authors
Line 53: “According to 5”.. please mention the authors of reference “5”.
Line 346: write in italic for species names.
Line 351: please mention the type of toxicity genes that are present.
Line 384: what method they used to identify the Sarcocystic species?
Line 411: please mention the source of study and what method they used for determination?.
Figure 6: what X and Y axis referring to?
Line 497: what kind of biotoxins does the author referring to?
The organization of the manuscript should be improved by following the author instructions
Author Response
Comments from reviewer 3.
Thank you very much for the review for improvements.
Line 53: “According to 5”.. please mention the authors of reference “5”.
Thank you for the comments. We revised the in-text citation. Please see line 98. Please see the revised manuscript.
Line 384: what method they used to identify the Sarcocystic species?
Please see the lines 470 and 471. The sample were identified by using morphological and molecular methods.
Line 346: write in italic for species names.
Thank you very much. Please see the line 372 of the revised manuscript.
Line 351: please mention the type of toxicity genes that are present.
The toxicity genes A and B are added. Please see lines 377and 378.
Line 411: please mention the source of study and what method they used for determination?
We added the relevant information as suggested. Please see the lines 441 and 442.
Figure 6: what X and Y axis referring to?
We added the captions for X and Y axis. Please see the figure under line 484 of the revised manuscript.
Line 497: what kind of biotoxins does the author referring to?
We referred paralytic shellfish toxin and tetrodotoxin. Please see line 541 and table 3.
The organization of the manuscript should be improved by following the author instructions
Thank you very much. We revised the manuscript as attached.

Round 2
Reviewer 2 Report
Comments and Suggestions for Authors
dear authors
the MS was improved, however, the plagiarism rate should be reduced
regards
Author Response
Comment: dear authors, the MS was improved, however, the plagiarism rate should be reduced. regards
Response: Thank you very much for reviewing our manuscript and insightful comments. Enclosed is a letter from Dr. Linda Nicolaides outlining the reasons for the high plagiarism rate. Additionally, we will provide the plagiarism check report as non-published material upon our re-submission The plagiarism check report was performed after we revised the manuscript based on the feedback received during the first round of the review.
Thank you once again for the comments which help us improve our manuscript.
